# Health data ecosystem in Pakistan: a multisectoral qualitative assessment of needs and opportunities

Sana Mahmood,[1,2,3] Ali Aahil Noorali ![ORCID],[4,5] Afshan Manji,[4,5] Noreen Afzal,[1] Saadia Abbas,[4] Javeria Bilal Qamar,[4] Sameen Siddiqi ![ORCID],[2] Zahra Hoodbhoy ![ORCID],[6] Salim S Virani,[4] Zulfiqar A Bhutta ![ORCID],[3,7] Zainab Samad ![ORCID] [4,5,8]

For numbered affiliations see end of article.

**Correspondence to**
Dr Zainab Samad;
samad.zainab@aku.edu

## ABSTRACT

**Objective** Data are essential for tracking and monitoring of progress on health-related sustainable development goals (SDGs). But the capacity to analyse subnational and granular data is limited in low and middle-income countries. Although Pakistan lags behind on achieving several health-related SDGs, its health information capacity is nascent. Through an exploratory qualitative approach, we aimed to understand the current landscape and perceptions on data in decision-making among stakeholders of the health data ecosystem in Pakistan.

**Design** We used an exploratory qualitative study design.

**Setting** This study was conducted at the Aga Khan University, Karachi, Pakistan.

**Participants** We conducted semistructured, in-depth interviews with multidisciplinary and multisectoral stakeholders from academia, hospital management, government, Non-governmental organisations and other relevant private entities till thematic saturation was achieved. Interviews were recorded and transcribed, followed by thematic analysis using NVivo.

**Results** Thematic analysis of 15 in-depth interviews revealed three major themes: (1) institutions are collecting data but face barriers to its effective utilisation for decision-making. These include lack of collection of needs-responsive data, lack of a gender/equity in data collection efforts, inadequate digitisation, data reliability and limited analytical ability; (2) there is openness and enthusiasm for sharing data for advancing health; however, multiple barriers hinder this including appropriate regulatory frameworks, platforms for sharing data, interoperability and defined win-win scenarios; (3) there is limited capacity in the area of both human capital and infrastructure, for being able to use data to advance health, but there is appetite to improve and invest in capacity in this area.

**Conclusions** Our study identified key areas of focus that can contribute to orient a national health data roadmap and ecosystem in Pakistan.

## INTRODUCTION

Data are essential for tracking and monitoring progress on health-related sustainable development goals (SDGs).[1–3] While data and data analytics are being used in high income countries to improve health equity, health

## STRENGTHS AND LIMITATIONS OF THIS STUDY

⇒ Our study participants were experts and decision-makers from multiple sectors, across provinces, and with work at the intersection of health and data science.
⇒ In-depth interviews with key informants allowed for a thorough exploration of the scope and challenges for health data science in Pakistan.
⇒ We did not conduct patient interviews to learn about their opinions about the application of health data science in Pakistan.

outcomes and continuously inform healthcare systems, their use in low and middle-income countries (LMICs) is lagging.[4–8] Investing in data ecosystems represents an important opportunity for monitoring and quickening progress on health-related SDGs in LMICs.[1 8–10]

With a population of 230 million, Pakistan, the fifth most populous LMIC, has a high estimated mortality and morbidity burden for various diseases, but its health system and health information system capacity is nascent.[11 12] However, during the COVID-19 pandemic, data were made nationally available in almost real-time, and data science methods were used to inform health policy and population-level interventions such as smart lock downs and vaccinations efforts.[13 14] Multistakeholder and interprovincial collaboration underpinned this successful effort and highlighted the need for a national health data ecosystem outside of crisis situations. To inform such future efforts, an understanding of the current perceptions around health data, its use in decision making and the health data ecosystem in Pakistan is required. To this end, we adopted a qualitative approach to understand the current landscape as well as perceptions on data in decision-making among a wide range of stakeholders.

## METHODS

### Study design and setting

This was an exploratory qualitative study with the primary objectives to comprehend the scope of the health data ecosystem in Pakistan, knowledge and attitudes around developing partnerships and sharing data, and perceptions around the need for developing health data science capacity in Pakistan.

The study was led by investigators at the Aga Khan University (AKU) in Pakistan. With a 40-year presence in Pakistan, AKU has well established partnerships at both provincial and national levels, with government and academia, enabling regular engagement in interdisciplinary policy discussions and fora.

### Study instrument

A semistructured interview guide was designed using carefully curated questions (available in online supplemental file 1). The guide prompted a detailed discussion on the landscape and scope of existing health data. Further discussion was rooted in potential facilitators and barriers to building a national health data collaborative that would contribute to improved health outcomes in Pakistan. This included understanding the nature of existing policies and collaboratives, the availability and need of human capital for health data initiatives, and structures—from governance to infrastructure, which were present or would need to be developed and implemented to allow for organisations across sectors to comfortably share data to advance health outcomes in Pakistan.

The guide was pilot tested among a diverse cohort of four individuals and judged for clarity of questions. Feedback from the pilot testing was incorporated to address gaps in the interview guide. Interviews were conducted by two female investigators (ZS and SM), who were the departmental chair and director, respectively, while the research staff (AAN, AAN, SA, JBQ) acted as observers. Standardisation was maintained across all interviews by ensuring that the same two interviewers conducted all the interviews with the same guide. Both interviewers had prior experience of conducting qualitative interviews. Each interview was conducted online for a duration that varied between 30 min and 2 hours.

### Sampling, inclusion and exclusion criteria

A scoping exercise was conducted to identify experts and relevant institutions. Through discussion, the investigators collectively identified key sectors in the health ecosystem of Pakistan for a landscape analysis, which formed the inclusion criteria: (1) university and academia with faculty in health and/or information technology (IT), (2) senior-level hospital management (both private and public), (3) government ministers (federal and provincial/state), (4) non-governmental organisations (NGOs) and (5) private-sector organisations (pharmaceutical, finance and health insurance sectors). There were no major or minor exclusion criteria.

Based on the SDGs 4, 5 and 9[15] (quality education, gender equality and industry, innovation and infrastructure, respectively), we performed a mapping of major institutions across these domains in the public and private health sectors, private organisations and NGOs. Major institutions were defined as those that were expected to have organisational maturity and scale in the area and capacity to collect data. Finally, individuals with at least 5 years of leadership experience in their respective domains were eligible to participate in the study.

Following convenience sampling to select key stakeholders with a particular focus on those with management/decision-making roles, invites were extended via email, and interviews were arranged. Thematic saturation was reached at 15 interviews, which comprised the final study sample. Figure 1 illustrates the study methodology.

### Data analysis

The six-step method of thematic analysis of Braun and Clarke guided the analytical process.[16] Interviews were first audio recorded and transcribed verbatim. Since the research staff were not only observers but also transcribers for the interviews, these roles helped them get familiarised and immersed with the collected data. Transcripts were imported into NVivo (V.12). An initial list of a priori codes was used by two research team members (AAN and AM) to code the transcripts. New codes emerging from the data, which were deemed relevant to the study objectives, were also added to this list. This codebook was refined through iteration and consensus among research staff, which helped in standardisation of the codes applied in all transcripts.

Data coded under similar codes were then grouped to identify major themes, which were paired with direct verbatim quotations from the interviewees. The themes were then reviewed to ensure adequate data and participant quotations supported the creation of each theme. All themes were then defined to convey an adequate description of its subthemes and relevant data. Finally, the results were written in a format to describe the analysed data in detail.

### Patient and public involvement

Patients and the public were not involved in the research design, analysis and dissemination of the findings.

### Ethical considerations

The study received approval from the Ethical Review Committee at AKU (ERC number 2021-5839-16883). Written informed consent over email for the study was obtained from each participant before starting the interview.

## RESULTS

We conducted 15 in-depth interviews with a diverse range of stakeholders from five centralised cohorts. Sector and designation of the participants are described in table 1.

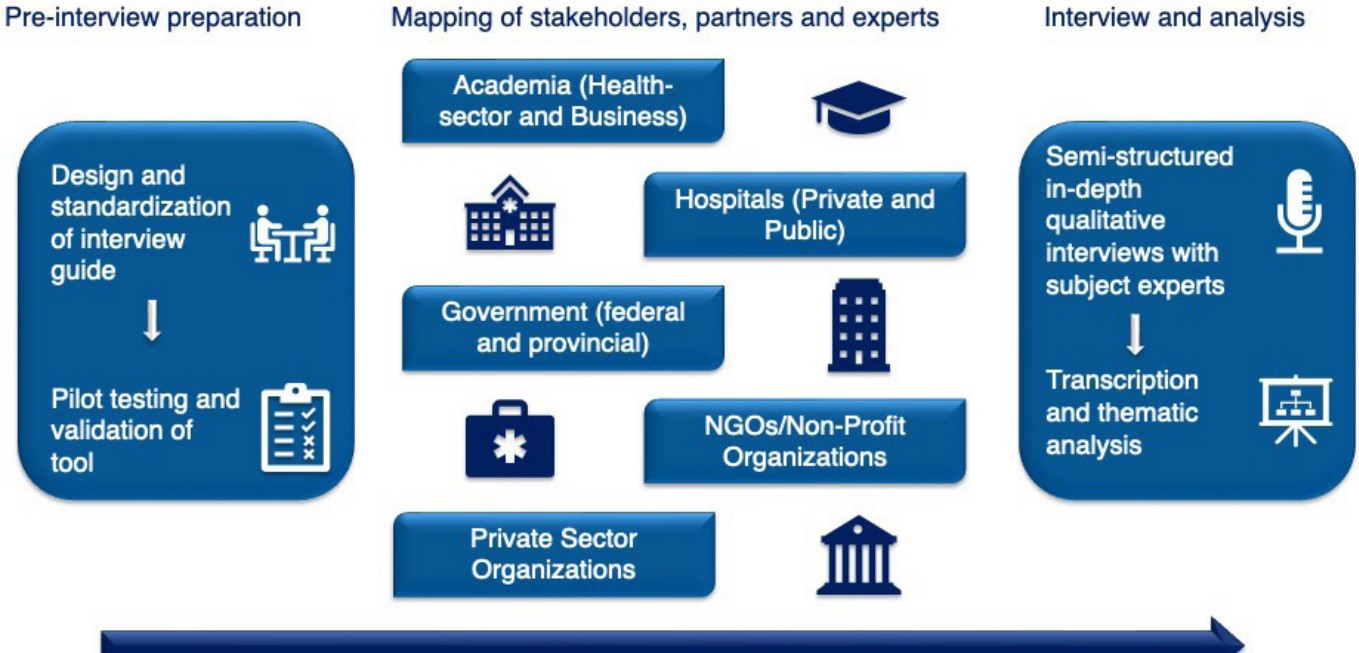

**Figure 1** An overview of the methodological framework of the studyparticipant cohorts, process of interview preparation, conductance and analysis.

Thematic analysis generated three overarching themes: (1) institutions are collecting data but face barriers to its effective utilisation for decision-making. These include lack of collection of needs-responsive data, lack of a gender/equity in data collection efforts, inadequate digitisation, data reliability and limited analytical ability; (2) there is openness and enthusiasm for sharing data for advancing health, however, multiple barriers hinder this endeavour; (3) there is limited capacity in the area of both human capital and infrastructure, for being able to use data to advance health, but there is appetite to improve capacity in this area.

### Theme 1: Institutions are collecting data but face barriers to its effective utilisation for decision-making. These include lack of collection of needs-responsive data, lack of gender/equity in data collection efforts, inadequate digitisation, data reliability and limited analytical ability

Experts communicated that there are several initiatives at the intersection of health and data, and organisations are collecting and holding data, but these initiatives exist in silos. Institutions have a large volume of operational data, but several barriers to their effective utilisation were identified. These ranged from lack of collection of needs-responsive data (data not capturing variables required for decision-making, interventions and policy reform), inadequate digitisation, inappropriate data formats, data reliability, value placed on ultimate data use by data collectors and human capacity to gauge scientific insights from it.

Inadequate data mapping and lack of data set awareness within institutions affecting access and use were identified as barriers. An expert in digital health strategy noted that *I would think that the dissemination of the data is a far bigger issue than the data holes being there. So, if you start digging, you find data sets, but you see that no one is aware of them, even though a lot of activity has happened.*

Leaders identified lack of needs-responsive data collection as a barrier to effective utilisation of data for decision-making. A provincial health minister stated that, *We find that getting population level information is really difficult. And you can get this information, but it's not really formalized— you just hear verbal estimates. So, in terms of planning, there is no common information base that people have that is like the gold standard.* In a similar vein, a leader in a prominent healthcare institution noted, *There are very few hospitals in*

| Table 1 | Sector and designation of study participants (n=15) | |
|---|---|
| **Sector** | **N (%)** |
| Academia | 1 |
| Hospitals | 2 |
| Government | 3 |
| Non-governmental organisations (NGOs) | 3 |
| Private-sector organisations | 6 |
| Designation | |
| Mid-level management | 5 |
| Chief medical officer | 2 |
| Health minister | 2 |
| Senior management | 6 |

*Pakistan that actually record quality and patient safety data. So, hospital quality level data, the kind that exists in the NHS, or in the US etc. doesn't exist here.* In addition, most participants noted that gender and equity lens have not been widely considered, neither during the collection and analysis of health data, nor in the design of research and data-driven initiatives. With regards to gender, a leader at a non-profit organisation explained that: *Disaggregated data by gender is probably a large problem nationally, particularly with development indicators.* Equity is often overlooked in conversations around data, and when considered, the level of commitment is inadequate. A governmental leader mentioned that: *Unfortunately, I think equity is more a function of conversations with development partners. And it does translate to some commitment, but not the level of commitment that should be the case.*

The digitisation and structure of available data are another holdup, as stated by a manager at an NGO: *Some organizations don't have data in a format that's easily accessible, because they don't have electronic systems.* Most participants shared the perception that there is a lack of an appropriately designed system to help collate data to its most desired format. This sentiment was shared by another expert in epidemiology at a tertiary care hospital, who noted that, *The reality was when I came in, I realized that the structure of the data being collected, was not conducive to be pulled out, right, so we couldn't do any research.* Another expert at an NGO noted, *When I joined X organization, all this information was collected on different platforms, some of this was paper based, some of it was collected with in-house applications. And none of it was necessarily standards-based. And none of these applications were well designed for information sharing.*

Data reliability was another aspect highlighted in the interviews. Specifically in the context of data sharing, a leader at an NGO noted, *You get conflicting information…. who is going to verify whether database A is correct or database B is correct.*

A provincial minister also suggested that while data-driven initiatives do exist, there is sometimes lack of clarity regarding the purpose and objectives of successfully collecting data and ultimately the value attached to these efforts among those collecting and managing this information. *We tried to give the lady health workers an app to update in real time to update the data about the diseases that they are seeing and the pregnant women that they are seeing. We don't have a proper accountability system and so we cannot tell people that this is something important for health policy and interventions and for a database for knowledge.*

Many interviewees felt that where data did exist, there were other obstacles like significant deficits in staffing and the inability to use the collected data. A leader in an NGO noted, *We do not have trained people in the system, even people who have dealt with data for a long time, do not have the analytical skills to make sense of it, draw conclusions, and ask questions. That has been a real challenge.*

## Theme 2: There is openness and enthusiasm for sharing data for advancing health, however, multiple barriers hinder this, including appropriate regulatory frameworks, platforms for sharing data, interoperability and defined win-win scenarios

Considering a shared vision to improve health outcomes in Pakistan, leaders indicated overall willingness to share data and partner for this common mission, expressing keen interest and stating their openness to proposals and collaborations. Defining win-win scenarios, in terms of shared objectives between entities and learning from each other's areas of expertise would be critical to organisations' sharing data, as a government leader observed: *I think we are open to proposals where, say, we make data in specific areas available. As long as we get something in return—if I can get more immediate impact out of that process, then that would excite me more. So, you name an area where we can problem-solve, and we can actually close the loop on that partnership (in terms of how we can actually translate it to some impact).* Interestingly, an epidemiologist at a tertiary care hospital shared insights about ensuring that a shared collaborative keeps 'democracy of data' as a central guiding principle: *I think there has to be democracy of data sharing within an organization because there's no point hanging on to data, and not sharing it so that somebody can make use of it, and that is one of the problems; people hang on to data as a good treasure that they cannot share with anybody.*

Most experts shared that certain challenges and pertinent questions would need to be accounted for to build a sustainable, shared, accessible data ecosystem. These include the validity and accuracy of data sets themselves, privacy and regulatory framework around data sharing, addressing systemic differences between different sectors and inadequate workforce and training. Experts shared that even if the data exist, dissemination of data is always an issue because data governance is a nascent field in Pakistan. The term governance is also used broadly by interviewees that use it to refer to privacy and security of data as well as an overarching governing structure to establish appropriate ownership of data. An entity leader, at a premier bank, noted that this also holds true for other sectors of Pakistan, such as the well-resourced finance sector, where one of the country's largest banks is still working through the early stages of data governance. Furthermore, structuring a data system to be useful requires understanding what data fields one has in their database, what should go into those data fields, and a system that is designed to ensure a clear utilitarian purpose. Though governance appears to be a major barrier for a data collaborative, subjects report that missingness and access to data are also impediments to utilisation.

A manager and planning executive at a bank, noted: *Some concerns in data sharing include who will own the data, what will be done with the data? Will the data remain valid or not, will transparency be maintained, privacy rules will be followed or not, ownership of the data? Where will data be used ultimately.*

Similarly, a chief medical officer at a tertiary hospital mentioned that: *There needs to be a lot more structure put into data sharing, and by structure, what I mean is that rules and regulations (which need to be set up a priori). That will really give confidence to individual institutions and individuals who own that data - that their data is going to be used properly, reliably and honestly.*

Interoperability of data sets was reported as a big challenge due to differences in data set formats and different data capacity/skills across different institutions and across public and private institutions. Issues of interoperability are further pronounced when complemented with differences in the approach of private versus public sectors and inpatient versus outpatient data. A leader in healthcare administration while describing the public sector healthcare stated that *Record keeping is something that is very poor there. In-patient record keeping is there, but there is nothing for out-patients at all. The private sector does keep the record, but the public sector does not. But the private sector does not share that data at all.*

### Theme 3: There is limited capacity in the area of both human capital and infrastructure, for being able to use data to advance health but there is appetite to improve and invest in capacity in this area

Inadequate human resources in data management and analytical skills in organisations were identified as a major barrier to both effective internal use of data and external collaborative data sharing efforts. Participants remarked that data sharing through a collaborative would require capacity-building in this area.

A manager at an NGO noted: *The issue is not whether people are willing to share data. But certain organizations, traditional non-profit ones for example, don't have data teams or data managers (because of cost budgetary constraints). So, I think that these organizations often don't have the capacity to manage data.* It was also mentioned that lack of capacity building was a barrier to successfully analysing and synthesising data, where many healthcare facilities at the district level were still using a paper-based format for recording data since their staff were not proficient in the use of technology. A provincial healthcare leader noted: *The main problem here is that we don't have an HR [human resource] there or a proper computerized system there to log in that data and upload it. We have a lot of restrictions in the IT department.* Similarly, a senior leader at a tertiary care hospital mentioned: *We have the data, but we don't have the capacity and capability to analyze it and make changes in healthcare.* There was a clear appetite and keen interest in investing organisational data capacity ranging from investing in needs-responsive data repositories and electronic systems, to upskilling the current workforce. A representative from academia shared their experience of setting up a data repository and its potential future impact: *The long-term objective of this Higher Education Data Repository initiative is that we collect all the granular information throughout a student's life cycle. This will generate a lot of data for analysis, about what kind of educational needs we have in our system, what kind of courses do we need to specialize in for our students and the areas that our faculty members need to specialize in.*

A representative from a large public sector tertiary care hospital reported their initiative of data automation and the barriers associated with it, notably, administrative and financial challenges around staffing plans: *We have begun an initiative at our hospital, where we are starting to do automation of the data and that is happening but I really wouldn't be able to say to what extent we have been successful with that. The data is within the automation center with the HR. That is also not working because our staffing there has been rejected.* This highlights the multifaceted nature of this dilemma—while some initiatives may be headed in the right direction, approvals to enact and sustain those initiatives are met with challenges.

An NGO leader described the process, components and importance of building a data-driven team—a cohesive unit of trained individuals that understand the application of data sciences to health. He stated: *There are four skills that I think are important in building out a data team that we found. One is data engineering, which is just someone who can query databases, particularly complex databases…Then, I think the next skill that we found useful is analytics, which is how to develop dashboards. And that is a very easily trainable skill. So, the third skill that's (commonly) not there is knowing what dashboard to make. And the fourth skill is data science, which is basically being able to model data and that's an even rarer skill especially locally.*

A key skill within building capacity to manage and analyse data is the ability to effectively communicate results. A government health leader also delineated the importance of communication in building health data capacity, which may be an essential, yet neglected, skill. After data collection and analysis, the success of any subsequent policy and prevention measures depended largely on how they are communicated to people. She stated that, *There must be a trigger factor that allows the person to do a communications strategy and awareness policy to send that message across. I think communication is very important. You specifically need to know how to communicate.*

A government leader also stated the need for and importance of on-the-job training by stating that *For me, the single silver bullet is creating data management routines that act as on the job training for managers.* He added, … *you need a data boot camp for health leaders, like that is weeklong. And that is sort of morning to night. And that's trying to basically break their thinking and get them to use differently the information that is available, because there is still a lot more information that is available.*

### DISCUSSION

Our study is the first, multidisciplinary endeavour to understand perceptions on health data and health data science in Pakistan. The main finding from our qualitative analysis is that the scope of data science in health for advancing health outcomes is far-reaching in Pakistan and likely in other LMICs where organisations have

collected a great deal of data but are in the early stages of understanding how best to leverage and use this data. Furthermore, there is potential for establishing a health data ecosystem comprised of a health data collaborative with an appropriate governance structure, intentionality toward data design elements focusing on gender, equity, and needs-responsiveness, that is supported by appropriate capacity building initiatives. Our study findings suggest that while we are at nascent stages of using data to progress a national health agenda, several independent and national efforts are being made to allow for digitisation and automation in healthcare and there is keen interest in investment in building capacity in this area. Even though the far-reaching scope of health data and data science methods in healthcare and their potential benefits are recognised by developing countries, development of a national data collaborative that might serve as a foundational block of a larger health data ecosystem is a complex endeavour and presents some challenges.[10] A principal lever for this agenda is timely access to the right information, but this has been a scarce resource in LMICs.[17]

Lack of rigorous and structured systems, problems with accuracy, credibility and completeness, inadequacy of trained personnel with core competencies, and unavailability of analytic tools were core obstacles highlighted by experts. Furthermore, there are very limited efforts to propagate a multimodal, multisectoral and multidisciplinary approach to data, leading to a conspicuous lack of a central repository of information in LMICs like Pakistan.[8] In line with the recent literature, a key concern highlighted by participants in data sharing was safeguarding their privacy, confidentiality and security, with all interviewees agreeing to the need of a governance and regulatory framework being set up a priori to ensure data transparency and maintaining the trust of all parties involved that their data will be used honestly and reliably.[8 18]

Our participants also stated that the analysis, design and collection of health data do not currently support gender and equity lens as the core of any organisation. Disaggregation of data by gender is a problem nationally and is difficult to find. Literature suggests that health system policy development does not always pay adequate attention to gender and even when policies do include gender, intentions can evaporate when it comes to actual implementation.[19 20] Study participants suggested that equity was often a function of conversation with development partners and that the level of commitment to inclusion of equity needed to be increased. Tannenbaum *et al* and The World Bank note that gender data is a powerful tool for improving lives as lack of disaggregated gender information has resulted in an incomplete disease understanding.[21 22] Furthermore, gender equity is an integral component of social responsibility and according to International Organization for Standardization 26000 SDG 5 (Guidance on Social Responsibility)*,* whereby the standard denotes the importance of having gender-inclusive

leadership and governance in ensuring elimination of gender bias and promotion of gender parity.[23]

Our study participants mentioned how organisations collect and store data, have information management systems, but that data are not being used in the most effective way, due to limited capacity and skillsets.[24] Hence, they emphasised that translation and evidence synthesis require significant capacity building. A systematic review reflects on the importance of ongoing training and multilevel strategies needed in development of such programmes, and how capacity building can influence different levels of entire organisations and systems.[25] The types of interventions assessed included internet-based teaching and workshops. The results of a worldwide cross-sectional survey by Kaggle *et al* illustrate the extent to which companies in various countries have adapted to machine learning models, with Israel surpassing even the USA.[26] This need also represents an opportunity to develop local, contextual health data science programmes that equip individuals with appropriate data management and data analytics skills.[27]

A national strategy on establishing a robust health data ecosystem and data collaborative for Pakistan will be an important next step. This necessitates that the gaps identified globally and in our qualitative interviews are bridged and data are put into action. In this regard, a national health digital framework has recently been developed by the Ministry of Health, which can be used for developing a high-level roadmap. As noted by healthcare experts, the roadmap is to help healthcare professionals use data science principles to inform decision-making, uplifting research and guiding clinical approaches to improve healthcare delivery.[28 29]

This study has a few limitations. Our findings mainly stemming from interviews among leaders in the healthcare system of Pakistan do not provide such larger room for generalisability beyond the Global South. However, we present perspectives from a low resource setting, which has contextual relevancy and implications for other LMICs in the region. Qualitative interviews focused on perspectives from key management leads at major institutions. This was primarily because the scope and objectives of this exercise were to assimilate input from experts and leaders in management, policy and healthcare. A next step, on establishing a health data collaborative, will be to ensure data and perspectives from patients and communities, who serve as key stakeholders in healthcare systems.

## CONCLUSION

The present study highlights important opportunities and barriers that need to be addressed to develop a health data ecosystem in Pakistan. Creation of appropriate governance, regulatory frameworks, gender and equity indicators, and defining win-win scenarios are important principles to consider for planning any national health data collaboratives. To enable this ecosystem, collaboration is required on strategic outlining of how data can

be collated, organised, curated, updated and finally pipelined. For achieving this goal, building data science capacity within organisations would be critical, thus providing the ability to leverage health data to its full potential for informed decision-making.

**Author affiliations**
[1]Dean's Office, Medical College, Aga Khan University, Karachi, Pakistan
[2]Department of Community Health Sciences, Aga Khan University, Karachi, Pakistan
[3]Institute of Global Health and Development, Aga Khan University, Karachi, Pakistan
[4]Department of Medicine, Medical College, Aga Khan University, Karachi, Pakistan
[5]Health Data Science Centre, Clinical and Translational Research Incubator, Medical College, Aga Khan University, Karachi, Pakistan
[6]Department of Pediatrics and Child Health, Medical College, Aga Khan University, Karachi, Pakistan
[7]Centre for Global Child Health, Hosp Sick Children, Toronto, Ontario, Canada
[8]Department of Medicine, Duke Clinical Research Institute, Durham, North Carolina, USA

**Contributors** All authors confirm that they had full access to all the data in the study and accept responsibility to submit for publication. ZS conceived of, designed the study, collected data and acquired funding. ZS is the guarantor of the study. SM collected the data and wrote the original draft of the manuscript. AAN analysed and interpreted the data and wrote the original draft of the manuscript. AM analysed and interpreted the data and wrote the original draft of the manuscript. NA, SS, ZH, ZAB, SV wrote the original draft of the manuscript. SA was involved in data curation and manuscript writing. JQB was involved in data curation and manuscript writing. All authors contributed to critically reviewing and editing the manuscript.

**Funding** This work was supported through a grant from the Bill & Melinda Gates Foundation (grant number: INV-021944).

**Competing interests** Dr ZS reports research grant support from the NIH-Fogarty International Center for the AKUPI-NCDs Research training program (Project Number 5D43TW011625-02) and UZIMA DS (U54TW012089); the NIHR, UK for the Centre for IMPACT (NIHR203248); Bill & Melinda Gates Foundation (INV 021944 & INV 050389); and Duke University. Dr SV MD, PhD, FACC, FAHA, FASPC Research support: U.S. Department of Veterans Affairs, National Institutes of Health, World Heart Federation, Tahir and Jooma Family; Honorarium: American College of Cardiology American College of Cardiology (Associate Editor for Innovations, acc. org); Associate Editor (Current Atherosclerosis Reports, Current Cardiology reports, Journal of Clinical Lipidology). Dr ZB MBBS, FRCPCH, FAAP, PhD Executive Director of NCD Child.

**Patient and public involvement** Patients and/or the public were not involved in the design, or conduct, or reporting, or dissemination plans of this research.

**Patient consent for publication** Not applicable.

**Ethics approval** The study received approval from the Ethical Review Committee at AKU (ERC number 2021-5839-16883). Participants gave informed consent to participate in the study before taking part.

**Provenance and peer review** Not commissioned; externally peer reviewed.

**Data availability statement** No data are available.

**ORCID iDs**
Ali Aahil Noorali http://orcid.org/0000-0002-5112-9571
Sameen Siddiqi http://orcid.org/0000-0001-8289-0964
Zahra Hoodbhoy http://orcid.org/0000-0002-0439-8293
Zulfiqar A Bhutta http://orcid.org/0000-0003-0637-599X
Zainab Samad http://orcid.org/0000-0003-2422-3199

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
