## [Reviewer comments · BMJ Open]

ARTICLE DETAILS

TITLE (PROVISIONAL)	Health Data Ecosystem in Pakistan – A Multi-sectoral Qualitative Assessment of Needs and Opportunities
AUTHORS	Samad, Zainab; Mahmood, Sana; Noorali, Ali Aahil; Manji, Afshan; Afzal, Noreen; Abbas, Saadia; Qamar, Javeria Bilal; Siddiqi, Sameen; Hoodbhoy, Zahra; Virani, Salim; Bhutta, Zulfiqar

VERSION 1 – REVIEW

REVIEWER	Possenti, Valentina National Centre for Disease Prevention and Health Promotion
REVIEW RETURNED	06-Feb-2023

GENERAL COMMENTS	The paper offers a pretty enjoyable and interesting reading as it builds on a clear structure and organisation. I would propose the revisions that follow as per the single sections. ABSTRACT The five major themes could be presented according to a more structured logic, given also that a couple of issues are repeated twice, such as the use of data (this applies also to the Results' organisation in the article body). In the Conclusions' line "inform" should be softened/downsized to something as "suggest the pathways to, contribute to orient" or similar. Conversely the sentence "Our findings might not be generalizable to high income countries" might become "... do not provide such larger room for generalizability...". Far from being only a formal matter, in the HICs some of the issues described in the paper apply. Of course, it is not the case for others. INTRODUCTION is plain and quite exhaustive, although some issues raised could be further exploited (e.g., it is interesting that COVID-19 represented somehow a globalised opportunity for public health growth to a certain extent. METHODS (Sampling, Inclusion and Exclusion Criteria), RESULTS In these parts, a better profiling of the study sample could be developed. Basing on recalling health-related SDGs (occurring just in the Introduction, but then the concept is not such that retrievable elsewhere in the paper), it is relevant to define that public health indicators are retrievable not only in the SDG 3 (namely, health-based) but also in other several SDGs and related targets. This is the focus of the first listed reference (GBD 2017), to this end the publication "Possenti V, Minardi V, Contoli B, Gallo R, Lana S, Bertozzi N, Campostrini S, Carrozzi G, Cristofori M, D'Argenzio A, De Luca AMC, Fateh-Moghadam P, Ramigni M, Trinito MO, Vasselli S, Masocco M. The two behavioural risk factor surveillances on the adult and elderly populations as information systems for leveraging data on health-related Sustainable Development Goals in Italy.
---

	International Journal of Medical Informatics; Volume 152, August 2021, 104443. DOI: 10.1016/j.ijmedinf.2021.104443” may also be helpful. It is possible to retrieve just the professional level (such as chief, manager, etc.) and the health (e.g., hospital) or financial affiliation (e.g., bank) of the 15 stakeholders interviewed. Table 1 reports these characteristics, which are not actually “demographic”, but in the text and table title should be changed simply in “Sector and designation...” as indicated in the table body. The cross-sectoral aspects of health data is quite crucial nowadays because, as described in many international strategies and plans (Agenda 2030 is one of them), health outcomes derive from a panel of sectors and institutions (education systems, environmental policies, housing planning, just to mention a few), and are managed accordingly. Then, Results could be improved as per their organisation and Discussion elaborated further accordingly. Limitations should be better defined but conversely the sentence on generalisability may be developed as indicated above. CONCLUSION also may be more precise more strictly linking up to the study outputs, with a clearer concept for terms such as “collaboration”.
--	--

REVIEWER	Dryden, Eileen M. VA Bedford Healthcare System Center for Healthcare Organization and Implementation Research
REVIEW RETURNED	13-Mar-2023

GENERAL COMMENTS	Overall I thought there was something interesting to be shared in this paper, but that the framing of it needs to be clearer, the methods need more detail, and the results and discussion need to be organized and tied together much better. Introduction  • I think a better case could be made for the need for this work – e.g. that the literature might suggest certain things (spell out) need to be in place for LMICs to begin to collect and use data more effectively as is done in higher income countries, we don’t know that this is true for Pakistan, therefore, we conducted this work. OR it is unclear from that literature what is needed, so it was essential to do this work. • A phrase like ‘digital data exhausts’ might be distracting to the reader. I would either explain further what you mean by that or use other, clearer terminology. • You discuss adopting “a qualitative approach to understand the current landscape and perception on data and in decision-making and health policy among a wide range of stakeholder.” And in the sentence prior to this call it ‘a mapping of national landscape’. In this manuscript I see that you gathered perceptions but not the mapping the landscape part as that terminology to me indicates a systematic documentation effort – very different than the thematic analysis that was done. If that was done but just not shared as part of this work, I would note that. If it wasn’t done, I would rephrase what your goals are. Methods  • Study Instrument –  o You indicate the interview guide was” judged for clarity of questions as well as face and content validity.” Face and content validity are concepts that are used with and more relevant to quantitative measures, not qualitative exploratory interview guides. I
---

don't think it is appropriate to use them here.

- o In the semi-structured guide attached, there aren't actual questions listed for Section 2 or 3 – just potential prompts in case 'of brief reply' – brief reply to what? It is unclear what was asked of the participants. If there was a clearer guide used, it should be attached.

- Sampling -
 - o I wonder if you were aiming to get a certain number of interviews per sector and if not, why not?
- Data analysis - More detail is needed in this section.
 - o Was the code book developed simply by reading the transcripts and coding what was interesting? That is what the section suggests. This would involve a very long process to come to agreement amongst a team to create a codebook in this manner. Generally, researchers come to the topic they are interested in with at least some a priori categories they are interested in knowing about and expect they might find due to the literature they are familiar with, these often are the topics of the questions in the interview guide. This is a place to start with creating a codebook. If you really did this totally inductively, I think you need to describe the process more – If nothing guided the initial code creation, how did that process work? If something did guide the process, what was it?
 - o Who was involved in the coding process (besides AAN and AM)?
 - o You note “this framework helped in the standardization of codes applied”. What framework? A framework is a term used in qualitative work and has a particular meaning. Do you just mean this process of meeting to come to consensus?
 - o Once coding was complete and you came to the analysis/interpretation/theme-making phase – who was involved in that?

Results

- Did anyone decline an invitation? If so, from which sector? Do you know why?
- The sector representation is quite uneven – did you try to make it even? If you tried to make it even but didn't achieve that, this could be added to limitation section. – unclear if you had more representation from academia if the results would have been different.
- You did a good job of transparency with the interviewers indicating they were female. Do you have any demographics for your study sample? Gender? Length of time in current position? Would be helpful to add if you have that.
- Theme 1
 - o You seem to be attributing a quote to 'The officials with the ministry of health' i.e. more than one person. Do you mean 'An official with the ministry of health'?
 - o Some of what you say seems contradictory – you discuss the 'lack of structured data' but then go on to say lower down “Although data are being rigorously collected there is lack of clarity....” Do you mean to say something more like “Even when data are rigorously collected there is lack of clarity....”
 - o To ease the burden on the reader I would keep like topics together. At the end of page 11 you discuss the structure of the available data being a barrier – it seems out of place there as this was the topic you discussed at the top of the page.
- Theme 2
 - o You discuss 'an expert' sharing insights about the democracy of data then go on to mention “an epidemiologist' with a quote discussing the democracy of data. Is this person one and the same or are two different people discussing this?

	Discussion  • I don't feel like the way the findings are portrayed at the beginning of the discussion are what you shared with us in the findings. The first finding 'that the scope of data science in health for advancing health outcomes, is far-reaching in Pakistan...' You seemed to demonstrate in the results that while a lot of data is being collected it is not being collected in a way that is systematic, rigorous or sharable. Secondly, you say there is potential for establishing a health data ecosystem....' Again, I don't think the way you framed your results you showed this. I think maybe there is enthusiasm for this and an understanding of the need for this, but it did not seem from the results there is a lot of potential for this yet. Then finally you go on to say the findings demonstrate "gender and equity must be intentionally included in the design of any collaborative" You might know that or feel that, but what you shared with the reader in the results simply says the people you spoke with note the data is not being collected in a way that would allow one to analyze by gender. You say this nicely later on when you do state what the results showed us: "Our participants also stated that the analysis, design and collection of health data does not currently support gender and equity lens as the core of any organization." • I think the discussion needs to be reworked to really place your findings in what you know of the literature to then go on and conclude what is needed. You seem to be conflating your findings and what you know is needed making it confusing. • There is a section dedicated to machine learning models. I think there is a little more explanation needed to tie this into the rest. Up to that point I thought capacity building and skills building was in building dashboards, querying databases, setting up rigorous systems – how does machine learning figure into this? Is this in place of training a cohort of data specialists or in addition to?
--	--

VERSION 1 – AUTHOR RESPONSE

Reviewer: 1

ABSTRACT The five major themes could be presented according to a more structured logic, given also that a couple of issues are repeated twice, such as the use of data (this applies also to the Results' organization in the article body). In the Conclusions' line "inform" should be softened/downsized to something as "suggest the pathways to, contribute to orient" or similar. Conversely the sentence "Our findings might not be generalizable to high income countries" might become "... do not provide such larger room for generalizability...". Far from being only a formal matter, in the HICs some of the issues described in the paper apply. Of course, it is not the case for others.

Authors' response:

Thank you for your valuable feedback. We have now rearranged the themes to give a more logical flow as follows:

1) Institutions are collecting data, but face barriers to its effective utilization for decision making. These include lack of collection of needs-responsive data, lack of a gender/equity in data collection efforts, inadequate digitization, data reliability, and limited analytical ability;

2) There is openness and enthusiasm for sharing data for advancing health, however, multiple barriers hinder this including appropriate regulatory frameworks, platforms for sharing data, inter-

operability, and defined win-win scenarios;

3) There is limited capacity in the area of both human capital and infrastructure, for being able to use data to advance health but there is appetite to improve and invest in capacity in this area

The conclusion has also now been modified as suggested:

Our study identified key areas of focus that can contribute to orient a national health data roadmap and ecosystem in Pakistan.

The study limitation has also been rephrased as follows:

Our findings may not provide room for generalizability, beyond the Global South.

INTRODUCTION is plain and quite exhaustive, although some issues raised could be further exploited (e.g., it is interesting that COVID-19 represented somehow a globalized opportunity for public health growth to a certain extent.

Authors' response:

Thank you for your valuable feedback. We have now modified the introduction section to incorporate the suggestions.

METHODS (Sampling, Inclusion and Exclusion Criteria), RESULTS In these parts, a better profiling of the study sample could be developed. Basing on recalling health-related SDGs (occurring just in the Introduction, but then the concept is not such that retrievable elsewhere in the paper), it is relevant to define that public health indicators are retrievable not only in the SDG 3 (namely, health-based) but also in other several SDGs and related targets. This is the focus of the first listed reference (GBD 2017), to this end the publication "Possenti V, Minardi V, Contoli B, Gallo R, Lana S, Bertozzi N, Campostrini S, Carrozzi G, Cristofori M, D'Argenzio A, De Luca AMC, Fateh-Moghadam P, Ramigni M, Trinito MO, Vasselli S, Masocco M. The two behavioural risk factor surveillances on the adult and elderly populations as information systems for leveraging data on health-related Sustainable Development Goals in Italy. International Journal of Medical Informatics; Volume 152, August 2021, 104443. DOI: 10.1016/j.ijmedinf.2021.104443" may also be helpful. It is possible to retrieve just the professional level (such as chief, manager, etc.) and the health (e.g., hospital) or financial affiliation (e.g., bank) of the 15 stakeholders interviewed. Table 1 reports these characteristics, which are not actually "demographic", but in the text and table title should be changed simply in "Sector and designation..." as indicated in the table body. The cross-sectoral aspects of health data is quite crucial nowadays because, as described in many international strategies and plans (Agenda 2030 is one of them), health outcomes derive from a panel of sectors and institutions (education systems, environmental policies, housing planning, just to mention a few), and are managed accordingly.

Authors' response:

Thank you for your comment.

We have also described the process of mapping our study participants, which was based on multiple SDGs as mentioned below:

Based on the SDGs 4, 5 and 9 (quality education, gender equality and industry, innovation and infrastructure respectively), we performed a mapping of major institutions across these domains in the public and private health sectors, private organizations and NGOs.

We have now renamed the title of the table to:

Table 1: Sector and designation of study participants (n=15)

Then, Results could be improved as per their organization and Discussion elaborated further accordingly. Limitations should be better defined but conversely the sentence on generalizability may be developed as indicated above.

Authors' response:

Thank you for your feedback and constructive critique. We have now rearranged the themes for them to flow in a more logical order as follows:

- 1) Institutions are collecting data, but face barriers to its effective utilization for decision making. These include lack of collection of needs-responsive data, lack of a gender/equity in data collection efforts, inadequate digitization, data reliability, and limited analytical ability;
- 2) There is openness and enthusiasm for sharing data for advancing health, however, multiple barriers hinder this including appropriate regulatory frameworks, platforms for sharing data, interoperability, and defined win-win scenarios;
- 3) There is limited capacity in the area of both human capital and infrastructure, for being able to use data to advance health but there is appetite to improve and invest in capacity in this area.

The limitation has also been rephrased as follows:

Our findings mainly stemming from interviews among leaders in the healthcare system of Pakistan do not provide such larger room for generalizability beyond the Global South.

CONCLUSION also may be more precise more strictly linking up to the study outputs, with a clearer concept for terms such as "collaboration".

Authors' response:

Thank you for your comment. We have now edited the conclusion as suggested:

The present study highlights important opportunities and barriers that need to be addressed to develop a health data ecosystem in Pakistan. Creation of appropriate governance, regulatory frameworks, gender and equity indicators, and defining win-win scenarios, are important principles to consider for planning any national health data collaboratives. To enable this ecosystem, collaboration is required on strategic outlining of how data can be collated, organized, curated, updated, and finally pipelined. For achieving this goal, building data science capacity within organizations would be critical, thus providing the ability to leverage health data to its full potential for informed decision making.

Reviewer: 2

Introduction

- I think a better case could be made for the need for this work – e.g. that the literature might suggest certain things (spell out) need to be in place for LMICs to begin to collect and use data more effectively as is done in higher income countries, we don't know that this is true for Pakistan, therefore, we conducted this work. OR it is unclear from that literature what is needed, so it was essential to do this work.
- A phrase like 'digital data exhausts' might be distracting to the reader. I would either explain further what you mean by that or use other, clearer terminology.
- You discuss adopting "a qualitative approach to understand the current landscape and perception on data and in decision-making and health policy among a wide range of stakeholder." And in the sentence prior to this call it 'a mapping of national landscape'. In this manuscript I see that you

gathered perceptions but not the mapping the landscape part as that terminology to me indicates a systematic documentation effort – very different than the thematic analysis that was done. If that was done but just not shared as part of this work, I would note that. If it wasn't done, I would rephrase what your goals are.

Authors' response:

Thank you for your comment. We have now modified the introduction section to incorporate the suggestions.

To avoid confusion, the phrase 'digital data exhausts' has been removed.

By mapping, we are only referring to the process of selecting our study participants from multiple sectors. We have now described this process in detail, whereby a careful review of the Sustainable Development Goals pertinent to the scope of our study was done to identify relevant stakeholders.

Methods

• Study Instrument –

o You indicate the interview guide was "judged for clarity of questions as well as face and content validity." Face and content validity are concepts that are used with and more relevant to quantitative measures, not qualitative exploratory interview guides. I don't think it is appropriate to use them here.

o In the semi-structured guide attached, there aren't actual questions listed for Section 2 or 3 – just potential prompts in case 'of brief reply' – brief reply to what? It is unclear what was asked of the participants. If there was a clearer guide used, it should be attached.

• Sampling -

o I wonder if you were aiming to get a certain number of interviews per sector and if not, why not?

• Data analysis - More detail is needed in this section.

o Was the code book developed simply by reading the transcripts and coding what was interesting? That is what the section suggests. This would involve a very long process to come to agreement amongst a team to create a codebook in this manner. Generally, researchers come to the topic they are interested in with at least some a priori categories they are interested in knowing about and expect they might find due to the literature they are familiar with, these often are the topics of the questions in the interview guide. This is a place to start with creating a codebook. If you really did this totally inductively, I think you need to describe the process more – If nothing guided the initial code creation, how did that process work? If something did guide the process, what was it?

o Who was involved in the coding process (besides AAN and AM)?

o You note "this framework helped in the standardization of codes applied". What framework? A framework is a term used in qualitative work and has a particular meaning. Do you just mean this process of meeting to come to consensus?

o Once coding was complete and you came to the analysis/interpretation/theme-making phase – who was involved in that?

Authors' response:

Thank you for your comment. We have now removed 'face and content validity' from the text.

For the semi-structured guide, prompts were listed for the convenience of the interviewers. In the interviews, these prompts were presented in the form of open-ended questions.

Our aim was to get representation from each key sector identified as relevant to the field of data science. The number of interviews was subject to Thematic saturation of the data. Since we were

looking at people in senior management roles specifically, their availability and scheduling issues posed difficulties in arranging more interviews.

We did have a priori categories guiding the analysis which are reflected in the overlapping themes in the interview guide and the final themes described in the results section. We have now described this under 'data analysis'.

For coding, SA and JBQ were also involved in the process.

We have now removed the word 'framework' to avoid confusion regarding the analytical process followed.

The principal investigator, who was also the interviewer, was consulted throughout the coding process and while developing themes to ensure data were being analyzed as accurately as possible.

Results

- Did anyone decline an invitation? If so, from which sector? Do you know why?
- The sector representation is quite uneven – did you try to make it even? If you tried to make it even but didn't achieve that, this could be added to limitation section. – unclear if you had more representation from academia if the results would have been different.
- You did a good job of transparency with the interviewers indicating they were female. Do you have any demographics for your study sample? Gender? Length of time in current position? Would be helpful to add if you have that.
- Theme 1
 - o You seem to be attributing a quote to 'The officials with the ministry of health' i.e. more than one person. Do you mean 'An official with the ministry of health'?
 - o Some of what you say seems contradictory – you discuss the 'lack of structured data' but then go on to say lower down "Although data are being rigorously collected there is lack of clarity...." Do you mean to say something more like "Even when data are rigorously collected there is lack of clarity...."
 - o To ease the burden on the reader I would keep like topics together. At the end of page 11 you discuss the structure of the available data being a barrier – it seems out of place there as this was the topic you discussed at the top of the page.
- Theme 2
 - o You discuss 'an expert' sharing insights about the democracy of data then go on to mention "an epidemiologist" with a quote discussing the democracy of data. Is this person one and the same or are two different people discussing this?

Authors' response:

None of the identified experts declined the interview invitation. However, as mentioned previously, scheduling issues limited the total number of people we were able to reach.

Secondly, a limited number of AMCs at this scale exists which is why we focused our attention on the largest AMC in the country.

We did not mention the gender of our study participants since some of them hold key positions in the government, and specifying their gender will reveal their identities and thus breach confidentiality.

We have now corrected the phrase to 'An official within the ministry of health'.

We have now reorganized the entire results section to describe the findings more clearly.

Yes, these phrases describe the same individual. Hence, we have now rephrased the sentence as

follows:

Interestingly, an epidemiologist at a tertiary care hospital shared insights about ensuring that a shared collaborative keeps 'democracy of data' as a central guiding principle.

Discussion

- I don't feel like the way the findings are portrayed at the beginning of the discussion are what you shared with us in the findings. The first finding "that the scope of data science in health for advancing health outcomes, is far-reaching in Pakistan..." You seemed to demonstrate in the results that while a lot of data is being collected it is not being collected in a way that is systematic, rigorous or sharable. Secondly, you say there is potential for establishing a health data ecosystem....' Again, I don't think the way you framed your results you showed this. I think maybe there is enthusiasm for this and an understanding of the need for this, but it did not seem from the results there is a lot of potential for this yet. Then finally you go on to say the findings demonstrate "gender and equity must be intentionally included in the design of any collaborative" You might know that or feel that, but what you shared with the reader in the results simply says the people you spoke with note the data is not being collected in a way that would allow one to analyze by gender. You say this nicely later on when you do state what the results showed us: "Our participants also stated that the analysis, design and collection of health data does not currently support gender and equity lens as the core of any organization."
- I think the discussion needs to be reworked to really place your findings in what you know of the literature to then go on and conclude what is needed. You seem to be conflating your findings and what you know is needed making it confusing.
- There is a section dedicated to machine learning models. I think there is a little more explanation needed to tie this into the rest. Up to that point I thought capacity building and skills building was in building dashboards, querying databases, setting up rigorous systems – how does machine learning figure into this? Is this in place of training a cohort of data specialists or in addition to?

Authors' response:

Thank you for your comment. To convey our findings more accurately, we have now rephrased the discussion to reflect the enthusiasm of our study participants as rightly pointed out by the reviewer. Our key informants were able to identify the current barriers to the use of data for building a health data ecosystem. Hence, our study provides a roadmap of all the things that need to be accounted for and focused on such as developing human capacity, training people etc. and so building on this can lead to the development of a health data ecosystem in the future.

With regards to the finding on gender, we have now reworded the sentence to describe the participants' verbatim and not allude to it as an interpretation.

With regards to the machine learning models, we have now placed this in the context of our findings and their implications for LMICs.

VERSION 2 – REVIEW

REVIEWER	Dryden, Eileen M. VA Bedford Healthcare System Center for Healthcare Organization and Implementation Research
REVIEW RETURNED	12-Jul-2023
GENERAL COMMENTS	Overall, this is resubmission is much improved from the first

	submission. There are some minor items that I believe still need to be addressed: 1. The entire manuscript needs to be reviewed carefully to improve the writing – e.g. ensure the first letter of a word at the beginning of sentences is capitalized, there are quotes at the end of quotations, and periods at the end of sentences. Special attention to the placement of commas is also needed.2. Ensure you are describing your results as intended - e.g. page 10. “Lack of adequate data awareness” is likely the barrier, not “adequate data awareness” which is what you have.3. You use the term ‘needs-responsive data’ throughout. Is that all data that is needed for decision-making? It is a little vague so I wasn’t quite sure. Adding a sentence to describe what you mean by that phrase would be helpful.4. I like the use of your quotes throughout the manuscript. There are times, though, where you let the quotes stand for themselves and don’t describe what the quote is illustrating. It would be helpful to briefly summarize what the quote illustrates so reader is not left wondering what the interviewee is getting at in the quote. See below for an example - it isn’t really clear what the interviewee is saying the barriers were... maybe lack of a particular type of staff?: A representative from a large public sector tertiary care hospital reported their initiative of data automation and the barriers associated with it: “We have begun an initiative at our hospital where we are starting to do automation of the data and that is happening ... That is also not working because our staffing there has been rejected so we are now trying to have a medical record system as an operative thing to see how the medical reforms will work.”
--	---

Reviewer: 2

1. The entire manuscript needs to be reviewed carefully to improve the writing – e.g., ensure the first letter of a word at the beginning of sentences is capitalized, there are quotes at the end of quotations, and periods at the end of sentences. Special attention to the placement of commas is also needed.
Author's response: Thank you for your comment. We have now thoroughly reviewed and edited the manuscript to improve the writing.

2. Ensure you are describing your results as intended -e.g. page 10. "Lack of adequate data awareness" is likely the barrier, not "adequate data awareness" which is what you have.
Author's response: Thank you for pointing this out. We have now corrected this as follows:
Inadequate data mapping and lack of dataset awareness within institutions affecting access and use were identified as barriers.

3. You use the term 'needs-responsive data' throughout. Is that all the data that is needed for decision-making? It is a little vague, so I wasn't quite sure. Adding a sentence to describe what you mean by that phrase would be helpful.
Thank you for your comment. We have now added the following description to describe this term:
These ranged from lack of collection of needs-responsive data (data not capturing variables required for decision making, interventions and policy reform), inadequate digitization, inappropriate data formats, data reliability, value placed on ultimate data use by data collectors, and human capacity to gauge scientific insights from it.

4. I like the use of your quotes throughout the manuscript. There are times, though, where you let the quotes stand for themselves and don't describe what the quote is illustrating. It would be helpful to briefly summarize what the quote illustrates so the reader is not left wondering what the interviewee is getting at in the quote. See below for an example - it isn't really clear what the interviewee is saying the barriers were... maybe lack of a particular type of staff?

A representative from a large public sector tertiary care hospital reported their initiative of data automation and the barriers associated with it: "We have begun an initiative at our hospital where we are starting to do automation of the data and that is happening ... That is also not working because our staffing there has been rejected so we are now trying to have a medical record system as an operative thing to see how the medical reforms will work."

Author's response: 4. I like the use of your quotes throughout the manuscript. There are times, though, where you let the quotes stand for themselves and don't describe what the quote is illustrating. It would be helpful to briefly summarize what the quote illustrates so the reader is not left wondering what the interviewee is getting at in the quote. See below for an example - it isn't really clear what the interviewee is saying the barriers were... maybe lack of a particular type of staff?

A representative from a large public sector tertiary care hospital reported their initiative of data automation and the barriers associated with it: "We have begun an initiative at our hospital where we are starting to do automation of the data and that is happening ... That is also not working because our staffing there has been rejected so we are now trying to have a medical record system as an operative thing to see how the medical reforms will work." Thank you for your comment.

We have added the full version of the quote to provide more context. We have also added a preceding and following sentence to help explain this more clearly. Additionally, we have added more context to four other quotes (below) that may help the reader understand the quotation more clearly.

A representative from a large public sector tertiary care hospital reported their initiative of data automation and the barriers associated with it, notably, administrative and financial challenges around staffing plans: " We have begun an initiative at our hospital, where we are starting to do automation of the data and that is happening but I really wouldn't be able to say to what extent we have been

successful with that. The data is within the automation center with the HR. That is also not working because our staffing there has been rejected”. This highlights the multifaceted nature of this dilemma – while some initiatives may be headed in the right direction, approvals to enact and sustain those initiatives are met with challenges.

It was also mentioned that lack of capacity building was a barrier to successfully analyzing and synthesizing data, where many healthcare facilities at the district level were still using a paper-based format for recording data since their staff were not proficient in the use of technology. A provincial healthcare leader noted: “The main problem here is that we don’t have an HR [human resource] there or a proper computerized system there to log in that data and upload it. We have a lot of restrictions in the IT department”.

A key skill within building capacity to manage and analyze data is the ability to effectively communicate results. A government health leader also delineated the importance of communication in building health data capacity which may be an essential, yet neglected, skill. After data collection and analysis, the success of any subsequent policy and prevention measures depended largely on how they are communicated to people. She stated that, “There must be a trigger factor that allows the person to do a communications strategy and awareness policy to send that message across. I think communication is very important. You specifically need to know how to communicate”.

A leader in healthcare administration while describing the public sector healthcare stated that “Record keeping is something that is very poor there. In-patient record keeping is there, but there is nothing for out-patients at all. The private sector does keep the record, but the public sector does not. But the private sector does not share that data at all”.

Defining win-win scenarios, in terms of shared objectives between entities and learning from each other’s areas of expertise would be critical to organizations sharing data, as a government leader observed: “I think we are open to proposals where, say, we make data in specific areas available. As long as we get something in return - if I can get more immediate impact out of that process, then that would excite me more. So, you name an area where we can problem-solve, and we can actually close the loop on that partnership (in terms of how we can actually translate it to some impact)”.

VERSION 3 – REVIEW

REVIEWER	Dryden, Eileen M. VA Bedford Healthcare System Center for Healthcare Organization and Implementation Research
REVIEW RETURNED	21-Aug-2023
GENERAL COMMENTS	This is a much improved version of your manuscript. There is one change I feel is still needed. Under the Data Analysis section, you start by saying you did "Grounded theory and the six-step method of thematic analysis of Braun and Clarke....". What you go on to describe, however, is not grounded theory and really, for the purposes of your work, grounded theory - generally considered a theory generating approach - would not have been appropriate. I would take that out and simply start with "The six-step method of Braun and Clarke...". This change is small but important so as not to confuse readers about the analysis approach you took.

VERSION 3 – AUTHOR RESPONSE

Reviewer 2 comment:

Under the Data Analysis section, you start by saying you did "Grounded theory and the six-step method of thematic analysis of Braun and Clarke....". What you go on to describe, however, is not grounded theory and really, for the purposes of your work, grounded theory - generally considered a theory generating approach - would not have been appropriate. I would take that out and simply start with "The six-step method of Braun and Clarke...". This change is small but important so as not to confuse readers about the analysis approach you took.

Author response:

Thank you for pointing this out. As suggested by the reviewer, we have now removed the phrase mentioning the grounded theory approach.